# The Comparative Analysis of Antiviral Activity of Native and Modified Fucoidans from Brown Algae *Fucus evanescens* In Vitro and In Vivo

**DOI:** 10.3390/md18040224

**Published:** 2020-04-22

**Authors:** Natalya V. Krylova, Svetlana P. Ermakova, Vyacheslav F. Lavrov, Irina A. Leneva, Galina G. Kompanets, Olga V. Iunikhina, Marina N. Nosik, Linna K. Ebralidze, Irina N. Falynskova, Artem S. Silchenko, Tatyana S. Zaporozhets

**Affiliations:** 1G.P. Somov Institute of Epidemiology and Microbiology, 690087 Vladivostok, Russia; galkom1965@gmail.com (G.G.K.); olga_iun@inbox.ru (O.V.I.); niiem_vl@mail.ru (T.S.Z.); 2G.B. Elyakov Pacific Institute of Bioorganic Chemistry, 690022 Vladivostok, Russia; swetlana_e@mail.ru (S.P.E.); artem.silchencko@yandex.ru (A.S.S.); 3I.I. Mechnikov Research Institute of Vaccines and Sera, 105064 Moscow, Russia; v.f.lavrov@inbox.ru (V.F.L.); wnyfd385@yandex.ru (I.A.L.); mnossik@yandex.ru (M.N.N.); lina.lidze@gmail.com (L.K.E.); falynskova@mail.ru (I.N.F.)

**Keywords:** fucoidans, HSV-1, HSV-2, ECHO-1, HIV, antiviral activity

## Abstract

The enzymatic depolymerization of fucoidans from brown algae allowed the production of their standardized derivatives with different biological activities. This work aimed to compare the antiviral activities of native (FeF) and modified with enzyme (FeHMP) fucoidans from *F. evanescens*. The cytotoxicity and antiviral activities of the FeF and FeHMP against herpes viruses (HSV-1, HSV-2), enterovirus (ECHO-1), and human immunodeficiency virus (HIV-1) in Vero and human MT-4 cell lines were examined by methylthiazolyltetrazolium bromide (MTT) and cytopathic effect (CPE) reduction assays, respectively. The efficacy of fucoidans in vivo was evaluated in the outbred mice model of vaginitis caused by HSV-2. We have shown that both FeF and FeHMP significantly inhibited virus-induced CPE in vitro and were more effective against HSV. FeF exhibited antiviral activity against HSV-2 with a selective index (SI) > 40, and FeHMP with SI ˃ 20, when they were added before virus infection or at the early stages of the HSV-2 lifecycle. Furthermore, in vivo studies showed that after intraperitoneal administration (10 mg/kg), both FeF and FeHMP protected mice from lethal intravaginal HSV-2 infection to approximately the same degree (44–56%). Thus, FeF and FeHMP have comparable potency against several DNA and RNA viruses, allowing us to consider the studied fucoidans as promising broad-spectrum antivirals.

## 1. Introduction

An increase in viral infections among all human infectious diseases and a lack of effective antivirals is one of the most significant problems of healthcare worldwide. Quite a few of currently used antivirals have a relatively low antiviral activity with a narrow spectrum of action (<one virus/ one medicine>), and viruses frequently become resistant to such antivirals. Therefore, the search for antiviral compounds that can effectively block the reproduction of a broad spectrum of human viruses and the development of new drugs based on such antivirals is quite essential.

Global pharmaceutical experience shows that marine hydrobionts have significant potential as raw materials for the development of drug products. In recent years, the sulfated polysaccharides from various marine algae, including fucoidans from brown algae, have attracted researchers’ attention. The main components of the fucoidans molecule are the sulfated L-fucose residues. In addition to fucose residues, fucoidans often contain other monosaccharides such as galactose, uronic acids, xylose, mannose and others [1]. These polysaccharides have various biological activities—anticoagulant, anti-inflammatory, antitumor, antioxidant, adjuvant, and antiviral [2,3,4,5]. Their biological properties are determined by the structure of the main chain, molecular weight, content, and location of sulfate and acetate groups [6,7,8,9]. A study of the pharmacokinetics and tissue distribution of fucoidans isolated from different types of brown algae showed that high-molecular-weight fucoidans are determined for a long time in the blood, mainly accumulating in the spleen, kidneys and liver, and excreted in the urine [10,11]. Fucoidans are the basis of biologically active compounds [12] used as drug carriers [13,14]. Therapeutic characteristics of fucoidans have been assessed in clinical trials [15,16]. However, despite the apparent progress in studying the properties of fucoidans, there are no registered fucoidans-based drug products, in part due to difficulties in obtaining standard polysaccharide samples.

One of the strategies for the development of simpler biologically active fucoidans with a standardized and reproducible structure is the enzymatic transformation of fucoidans [17,18]. Previously, the enzymatic depolymerization of the native fucoidan from brown alga *Fucus evanescens* was carried out using recombinant fucoidanase FFA1 from marine bacteria *Formosa algae* to obtain a standardized fragment of fucoidan molecule with the regular structure [19]. In the present study, we used both fucoidans:—native fucoidan isolated from *F. evanescens* — FeF (*F. evanescens* Fucoidan) and its derivative obtained after enzymatic treatment — FeHMP (*F. evanescens* High-Molecular-Weight Product). The FeF (molecular weight 160 kDa), as previously shown, built up from randomly alternating α-(1→3)- and α-(1→4)-linked residues of sulfated fucose residues [20,21,22]. The FeHMP (molecular weight—50.8 kDa) has a regular structure and built from repeating fragments of ([→3)-α-L-Fucp(2,4OSO_3_−)-(1→4)-α-L-Fucp(2OSO_3_−)-(1→]_n_) [19,20,21,22].

The aim of this study was the comparative analysis of the antiviral activity of the native fucoidan from *F. evanescens* (FeF) and its derivative (FeHMP) with the regular structure against several DNA- and RNA-containing viruses. Investigation of the antiviral activity of native fucoidan (FeF) with nonregular structure and standardized fucoidan (FeHMP) can give us the answer of which structural fragment of these unique polysaccharides is important against viruses.

## 2. Results

### 2.1. The Cytotoxicity and In Vitro Antiviral Activity of Fucoidans against DNA and RNA Viruses

The data for cytotoxicity determined by MTT (methylthiazolyltetrazolium bromide) assay showed low toxicity of the native (FeF) and modified (FeHMP) fucoidans and Acyclovir^®^ against Vero cells—their 50% cytotoxic concentrations (CC_50_) were above 2000 μg/mL, while CC_50_ of Ribavirin^®^ was 750 μg/mL. Antiviral activity assay was performed at the concentrations below 500 µg/mL. In the case of MT-4 cells, the CC_50_ of fucoidans were 200 μg/mL, and CC_50_ of the Retrovir^®^ and Epivir^®^ were 50 μg/mL. Thus, their anti-HIV-1 activity was assessed at the concentrations below their CC_50_.

The antiviral effect of fucoidans against herpes simplex viruses type 1 and 2 (HSV-1 and HSV-2), enterovirus (ECHO-1) and human immunodeficiency virus (HIV-1) was assessed using cytopathic effect (CPE) inhibition assay. To study the inhibitory effect of tested compounds on the stage of virus infection, the fucoidans were added before virus infection (pretreatment of cells), directly to the virus suspension (pretreatment of virus), concurrently with the initiation of virus infection (simultaneous treatment), and after penetration of the virus into host cells (treatment of infected cells), respectively. The results of the virus-inhibitory activity of tested compounds were used for calculations of the 50% inhibitory concentration (IC_50_) and the selectivity index (SI) for each of the compounds (Table 1).

The pretreatment of DNA-containing viruses (HSV-1 and HSV-2) with fucoidans (direct virucidal action) showed a moderate antiviral activity of FeF and FeHMP (their mean values of SI were ~ 19). In the case of RNA-containing viruses, the virucidal action of fucoidans against HIV-1 was modest (SI = 8), and minor against nonenveloped RNA-containing ECHO-1 virus (SI~3). This method of application of fucoidans showed no significant difference between FeF and FeHMP (*p* ˃ 0.05) (Table 1).

The treatment of cells with fucoidans before infection (preventive effect) and just after the virus inoculation (0 h) (simultaneous treatment) revealed the highest antiviral activity of tested compounds (Table 1). The native fucoidan effectively inhibited the replication of both HSV types when compared with modified fucoidan—SI of FeF was 1.5–1.9 times higher than with FeHMP. In the case of ECHO-1, the antiviral activity of fucoidans was moderate (SI~22), and the difference between the antiviral effect of FeF and FeHMP was not significant (*p* ˃ 0.05). The anti-HIV-1 activity of fucoidans in this application mode was modest (SI~8).

The application of sulfated polysaccharides after virus adsorption and penetration to cells (at 1 h postinfection) (treatment of infected cells) showed moderate replication inhibition against HSV-1, HSV-2, and ECHO-1 (the average SI was 23) and modest virus-inhibitory activity of fucoidans against HIV-1 (SI~8). The difference between FeF and FeHMP was not significant (*p* ˃ 0.05) (Table 1).

Our results demonstrated the ability of native and modified fucoidans to inhibit the replication of HSV-1, HSV-2, ECHO-1, and HIV-1; herpes simplex viruses, especially HSV-2, were the most sensitive to tested sulfated polysaccharides.

### 2.2. In Vivo Efficacy of Fucoidans against HSV-2 Infection in a Mouse Vaginitis Model

Having identified the antiviral potency of native and modified fucoidans in vitro, we evaluated their protective efficacy against the intravaginal HSV-2 challenge in mice. The clinical symptoms of infection were observed to start on the 5th day postinfection and included loss of body weight, vaginal swelling, hyperemia, and discharge. Also, the decreased motor activity, food, and water intake were observed, followed with hind limb paralysis on the 7th day postinfection. The average survival time of infected animals in the virus group was 9.7 ± 2.6 days (Table 2).

The treatment of HSV-2-infected mice with fucoidans from *F. evanescens* (FeF and FeHMP) resulted in a dose-dependent antiviral effect, leading to reduced clinical symptoms and mortality (Table 2, Figure 1A). Fucoidans (FeF and FeHMP) given intraperitoneally at 10 mg/kg/d provided a survival rate of 44–56% in infected mice. The average survival time of fucoidans-treated mice increased by 4–6 days compared to the virus group (*p* ≤ 0.05). Furthermore, the administration of these doses of fucoidans to HSV-2-infected animals effectively prevented the loss of their body weight (Figure 1B). The treatment of infected mice with Acyclovir^®^ (50 mg/kg/d) protected all animals from the lethal outcome, so the protection index was 100%.

On the 7th day postinfection, the replication of HSV-2 in the vaginal epithelium of mice in the virus group reached 3.63 ± 0.18 lg TCID_50_/mL (Figure 1C). The intraperitoneal administration of fucoidans at 10 mg/kg/d significantly reduced virus replication in the infected vagina compared to the virus group (*p* ≤ 0.05) (Figure 1C). By the seventh day of the treatment with 10 mg/kg/d of native fucoidan (FeF), a reduction of HSV-2 titers in the vaginal lavages by 2 lg TCID_50_/mL was observed. Similarly, the modified fucoidan (FeHMP, 10 mg/kg/d) and Acyclovir^®^ (50 mg/kg/d) significantly reduced HSV-2 titers by 3 lg TCID_50_/mL. Based on these data, it can be concluded that fucoidans FeF and FeHMP effectively protect mice from intravaginal HSV-2 challenge.

## 3. Discussion

It is well-known that fucoidans from different brown algae have antiviral activity against DNA-containing [24,25,26] and RNA viruses [27,28,29,30], and are promising candidates for the development of fucoidan-based drug products. In the meantime, the standardization of fucoidan extraction is a complicated process [31,32], mainly due to the structural diversity of fucoidans [33] and sometimes because of biotechnological problems [34]. Therefore, the critical task is the identification of the structural fucoidans fragments responsible for their biological activity as well as the development of standardized fucoidan-based drug products with a pronounced antiviral effect.

Our study aimed to investigate the effect of the native fucoidan and its regular derivative at the different stages of the virus life cycle, including significant human pathogens HSV-1, HSV-2, HIV-1, and ECHO-1. The results showed the ability of both fucoidans to increase the resistance to virus infection (preventive effect), directly affect virus particles (virucidal effect), and inhibit the early stage of virus replication (virus-inhibiting effect). The multifaceted mechanisms of action of tested fucoidans are similar to the mechanisms of action of other sulfated polysaccharides from different algae [35,36]. Nevertheless, we showed some specific features of the antiviral activity of FeF and FeHMP against the aforementioned viruses. For example, tested fucoidans more effectively inhibited the replication of the DNA-containing HSV-1 and HSV-2, compared with a moderate antiviral effect against RNA viruses ECHO-1 and HIV-1. It was found that fucoidans affected different stages of HSV-1 and HSV-2 replication more effectively at the stage of virus adsorption and penetration to the host cells, as demonstrated by the treatment of cells with fucoidans before virus infection and simultaneous treatment of cells with the mixture of fucoidan and virus. We hypothesize that one of the possible mechanisms of high anti-HSV activity of studied fucoidans can be related to the sulfated polysaccharides’ ability to interact with cells to competitively inhibit binding sites usually used by herpesviruses for cell entry [35,37,38]. 

Comparative analysis of antiviral activities of native (FeF) and modified (FeHMP) fucoidans showed that FeF more effectively inhibits the replication of both types of HSV than FeHMP. On the other hand, both tested fucoidans had similar antiviral activity against ECHO-1 and HIV-1. It was previously demonstrated [19] that FeHMP had lower molecular weight and was significantly more sulfated than FeF. Some authors reported that the antiviral activity of sulfated polysaccharides increased with molecular weight and sulfate content [37,38,39]. However, we believe that the revealed differences in the anti-HSV activity of FeF and FeHMP can be associated with the fine structure of both polysaccharides rather than with a decrease in the molecular weight or increase in the number of sulfate groups in FeHMP fucoidan.

It is possible that enzymatic hydrolysis brings about the decrease of structural fragments’ diversity in the fucoidan molecule, leading to the reduction in the number of potential fucoidan targets on the cell surface. Presumably, it leads to the lower anti-HSV activity of the modified fucoidan FeHMP compared with the native fucoidan FeF.

In our opinion, the antiviral activity of tested fucoidans from *F. evanescens* (FeF and FeHMP) against nonenveloped RNA viruses (for example, ECHO-1 enterovirus), is very important. Currently, there are only a few reports concerning the antienteroviral activity of sulfated polysaccharides from marine algae [40,41,42]. Our results suggest that mechanisms of anti-ECHO-1 activity of fucoidans can be related to the inhibition of virus adsorption and early stages of viral infection. It should be noted that in the case of enterovirus, the fucoidans antiviral effect was greater than that of Ribavirin^®^ (Table 1).

The last ten years were characterized by the appearance of numerous reports about the anti-HIV activity of sulfated polysaccharides isolated from different algae [43,44,45]. It is believed that the primary mechanism of anti-HIV action is the ability of these compounds to block the virus penetration to the sensitive host cells via binding to positively charged amino acids of virus envelope glycoprotein gp120, which mediates viral attachment [4,38,46]. Several authors reported inhibition of the HIV-1 life cycle both before and after virus penetration to cells with some sulfated polysaccharides [47,48]. Our study results also showed that native and modified fucoidans inhibited different stages of HIV-1 replication in human T cells (MT-4 cell line) when applied at the same inhibitory concentration (IC_50_ = 25 μg/mL) (Table 1).

The high worldwide prevalence of HSV-2 infections, the severity of complications and their close linkage with cervical cancer and HIV-1 infection allow attributing the diseases caused by HSV-2 to global healthcare and social issues [49,50,51]. Therefore, the search for anti-HSV-2 drugs with different mechanisms of action is still very important. Previously, the anti-HSV-2 activity of sulfated polysaccharides from seaweed and mushroom was shown in the mouse model of genital herpesvirus infection [52,53]. In our study, the protective efficacy of tested fucoidans against intravaginal HSV-2 challenges in mice was demonstrated (Table 2, Figure 1). Intraperitoneal administration of native fucoidan FeF and its derivative FeHMP significantly improved the survival rate, alleviated symptoms of the disease, prevented the weight loss, and reduced vaginal virus load induced by HSV-2 infection compared to the virus group. We assume that the protective antiviral effect of fucoidans is associated not only with selective influence on a different stage of viral infection but also with antioxidant, anti-inflammatory, and immunomodulatory properties of these compounds.

The results of our study showed that standardized fucoidan with a regular structure as well as native fucoidan have comparable potency against a range of DNA- and RNA-containing viruses associated with severe human pathology. We suggest that the antiviral properties of these unique polysaccharides may be due to structural features, and in particular, with their highly sulfated fragment ([→3)-α-L-Fucp(2,4OSO_3_−)-(1→4)-α-L-Fucp(2OSO_3_−)-(1→])_n_. Obtaining standardized fucoidan preparations can be a basis of a successful strategy for the development of promising broad-spectrum antivirals.

## 4. Materials and Methods

### 4.1. Viruses and Cell Cultures

The following viruses were used for the study—herpes simplex virus type 1 (HSV-1) strain VR3 was obtained from the National Collection of US Viruses (Rockville, MD, USA). The herpes simplex virus type 2 (HSV-2) strain G ATCC VR-734 was obtained from the Smorodintsev Research Institute of Influenza (Sankt-Petersburg, Russia). The strain IP91 of ECHO-1 enterovirus was obtained from the Chumakov Federal Scientific Center for Research and Development of Immune and Biological Products (Moscow, Russia). The human immunodeficiency virus (HIV-1) strain M070, subtype A6, was from the collection of HIV strains of the Mechnikov Research Institute of Vaccines and Sera (Moscow, Russia).

HSV-1, HSV-2, and ECHO-1 were grown in African green monkey kidney (Vero) cells using Dulbecco’s Modified Eagle’s Medium (DMEM) supplemented with 10% Fetal Bovine Sera (FBS) and 100 U/mL of gentamycin at 37 °C in a CO_2_ incubator; in the maintenance medium, the FBS concentration was decreased to 1%. HIV-1 was propagated in the human T-cell line MT-4 using RPMI-1640 medium supplemented with 10% FBS, 0.06% L-Glutamine, and 50 U/mL gentamycin.

### 4.2. Animals

Female outbred mice (16–20 g) were obtained from the Scientific Center for Biomedical Technology of the Federal Medical and Biological Agency (Andreevka, Moscow Region, Russia) and used for in vivo experiments. All procedures were performed strictly following the (European Convention for the Protection of Vertebrate Animals Used for Experimental and other Scientific Purposes) of 18 March 1986. All animal experiments comply with bioethical standards; the study protocol was approved by the local institutional bioethics committee of the Mechnikov Institute (protocol N 4, of 20.08.2019).

### 4.3. Studied Compounds 

Native fucoidan (FeF)—sulfated polysaccharide from brown algae *Fucus evanescens*.

Fucoidan from *F. evanescens* (FeF) was purified as described early [54] and characterized [20,21,22]. According to NMR spectroscopy data, native fucoidan FeF was built up from alternating α-(1→3)- and α-(1→4)-linked residues of sulfated fucose residues: (→3)-α-L-Fucp(2,4OSO_3_−)-(1→4)-α-L-Fucp(2OSO_3_−)-(1→) and (→3)-α-L-Fucp(2OSO_3_−)-(1→4)-α-L-Fucp(2OSO_3_−)-(1→). Modified fucoidan (FeHMP)—high molecular weight product (HMP) of enzyme hydrolysis of native fucoidan.

To obtain a fucoidan derivative with a regular structure, we used the method described in [19]. In brief, fucoidan from *F. evanescens* (FeF, 0.5 g) was dissolved in 49 mL of the buffer (0.04 M Tris-HCl, pH 7.0 with 5 mM CaCl_2_) and 1 ml of enzyme (fucoidanase FFA1, 0.1 mg/mL) was added. The reaction mixture was incubated at 34 °C for 72 h, and then heated at 80 °C for 10 min, and the precipitate was removed by centrifugation. The high-molecular-weight reaction products (FeHMP) (in supernatant) were precipitated with ethanol at the ratio of 1:3 (v/v), and precipitate was separated by centrifugation at 10,000× *g* for 40 min. ^1^H NMR spectra of both native and modified fucoidans are presented in Appendix A. FeHMP has a regular structure and consists of a repeating fragment: ([→3)-α-L-Fucp(2,4OSO_3_−)-(1→4)-α-L-Fucp(2,4OSO_3_−)-(1→])_n_ [19].

Summarized information with structural characteristics of fucoidans is presented in Table 3.

Fucoidans (FeF and FeHMP) and Acyclovir^®^ for experiments in vitro and *in vivo* were diluted in DMEM. Stock solution (10 mg/mL) of Ribavirin^®^ for in vitro experiments was dissolved in dimethylsulfoxide (DMSO, Sigma, Saint-Louis, MO, USA), stored at −20 °C and was diluted with a suitable cell culture medium to a final concentration of 0.5% DMSO. Antiretrovirals were diluted with RPMI-1640 (Pan-Eco, Moscow, Russia).

### 4.4. Cytotoxicity Assay of the Fucoidans

The cytotoxicity of fucoidans was estimated by MTT assay in Vero cells [55]. Cell monolayers (1 × 10^4^ cells/well) in the 96-well plates were treated with different concentrations of compounds (from 0.2 to 2000 μg/mL) and incubated at 37 °C in a CO_2_-incubator for three days; untreated cells were used as controls. MTT solution (methylthiazolyltetrazolium bromide, Sigma, Saint-Louis, MO, USA) was added to cells in the concentration 5 mg/mL, following incubation for 2 h at 37 °C. Then, the MTT solution was removed, and isopropanol was added to dissolve the insoluble formazan crystals. The optical density was measured at 540 nm using an ELISA microplate reader (Labsystems Multiskan RC, Vantaa, Finland). Cytotoxicity was expressed as 50% cytotoxic concentration (CC_50_) of each studied compound that reduced the viability of treated cells by 50% compared with untreated cells and was calculated using regression analysis of dose-response curve [56].

The cytotoxicity of fucoidans for lymphoblast cell line MT-4 was also assessed by an MTT assay. The cells were incubated in the 96-well plates, treated with different concentrations of fucoidans (0.25–250 μg/mL), and cultured at 37 °C in a CO_2_ incubator for five days; the untreated MT-4 cells were used as controls.

### 4.5. Antiviral Activity Assay of Fucoidans In Vitro

The antiviral activity of fucoidans against HSV-1, HSV-2, ECHO-1, and HIV-1 was evaluated using the cytopathic effect (CPE) inhibition assay in the Vero and MT-4 cells, respectively. The monolayers of Vero cells grown on 96-well plates (1 × 10^4^ cells/well) were infected with 100 TCID_50_/mL (the 50% tissue culture infectious dose) of the corresponding virus (HSV-1, HSV-2, ECHO-1). Suspension of MT-4 cells (3 × 10^5^cells/mL) was cultured in 96-well flat-bottomed plates followed by infection of 100 TCID_50_/mL of HIV-1. Some schemes of fucoidans and appropriate pharmaceutical application were investigated; each of them was carried out in three independent replicates in triplicate, with different concentrations of compounds (0.25–250 μg/mL). The plates were incubated at 37 °C in a CO_2_ incubator for 72 h for HSV-1, HSV-2, and ECHO-1 and 120 h for HIV-1 until 80–90% CPE was observed in virus control compared with cell control.

The pretreatment of cells with fucoidans. The monolayer of cells was pretreated with different concentrations of studied compounds for 2 h at 37 °C. After washing, the cells were infected with 100 TCID_50_/mL of the virus at 37 °C for one hour. Then, unabsorbed virus was removed by washing with phosphate-buffered saline (PBS), and cells were incubated in the maintenance medium until CPE appeared.

The pretreatment of the virus with fucoidans. An infectious dose of virus (100 TCID_50_/mL) was mixed with different concentrations of studied compounds at a ratio 1:1 (v/v), incubated for one hour at 37 °C; then, the mixture was applied to the monolayer of cells. After 1 h adsorption at 37 °C, the cells were washed with PBS, overlaid by the maintenance medium, and followed by incubation until CPE was observed. 

The simultaneous treatment of the cells with fucoidans and viruses. The monolayer of cells was infected with the virus (100 TCID_50_/mL) and simultaneously treated by different concentrations of the studied compounds (virus: compound, 1:1 v/v) for one hour at 37 °C. After virus adsorption, the mixture was removed; the cells were washed with PBS and incubated in maintenance medium until CPE appeared.

The treatment of virus-infected cells with fucoidans. The monolayer of cells was infected with the virus (100 TCID_50_/mL) at 37 °C for 1 h, and then the cells were washed with PBS and treated with different concentrations of the studied compounds and incubated before apparent CPE.

After the incubation, the cell supernatants were collected, virus titers were calculated using the Reed-Muench method [57] and expressed as TCID_50_/mL. The antiviral effect of fucoidans was determined by the difference of viral titers between the treated infected cells and untreated infected cells and expressed as virus inhibition rate (IR, %). IR was calculated according to the following formula [58]: IR = (1 − *T*/*C*) × 100%, where *T* is the antilog of the compound-treated viral titers, and C is the antilog of the control (without compound) viral titers. The fifty per cent inhibitory concentration (IC_50_) of each compound was determined as the compound concentration that inhibited virus-mediated CPE by 50% and was calculated using a regression analysis of the dose-response curve [56]. Selectivity index (SI) was calculated as the ratio of CC_50_ to IC_50_ for each compound.

### 4.6. Antiviral Activity Assay of Fucoidans against HSV-2 In Vivo

To evaluate the anti-HSV-2 activity of fucoidans *in vivo*, we used a mouse vaginitis model by vaginal infection with the HSV-2. Female outbred mice (16–20 g) were randomly divided into 7 groups with 10 mice in each group as follows—normal group (control uninfected mice), virus group (infected and untreated mice), and infected and treated groups, including FeF-treated (5 mg/kg/d and 10 mg/kg/d), FeHMP-treated (5 mg/kg/d and 10 mg/kg/d), and Acyclovir-treated (50 mg/kg/d). The mice in the normal and virus groups were administered saline.

Genital herpes virus infection of mice was established by vaginal inoculation of 30 μL HSV-2 (10^5^ TCID_50_/mL). At 1 h postinoculation and subsequently once a day for five consecutive days, all tested compounds were administered intraperitoneally, and the animals were observed daily for 21 days to calculate the survival rate, death protection, average survival time, and body weight change.

Also, the change of virus titer was analyzed—vaginal lavages were collected from the animals in each group on the 5^th^ and 7^th^ days postinfection. Briefly, 30 μL precooled DMEM was introduced to the vagina of infected mice and pipetted, collected vaginal lavages were adjusted to 150 μL by DMEM and centrifuged (500 g, 10 min). Following this, 10-fold dilutions of supernatants were placed on Vero cells monolayer in 96-well plates, incubated for three days at 37 °C in a CO_2_-incubator, and virus titer (lg TCID_50_/mL) was determined by the Reed-Muench method [57].

### 4.7. Statistical Analysis

Statistical analysis was done with Statistica 10.0 software (StatSoft Inc, Tulsa, OK, USA). The results are given as mean ± standard deviation (SD). The differences between parameters of control and experimental groups were estimated using the Wilcoxon test. Differences were considered significant at *p* ≤ 0.05.

## Figures and Tables

**Figure 1 marinedrugs-18-00224-f001:**
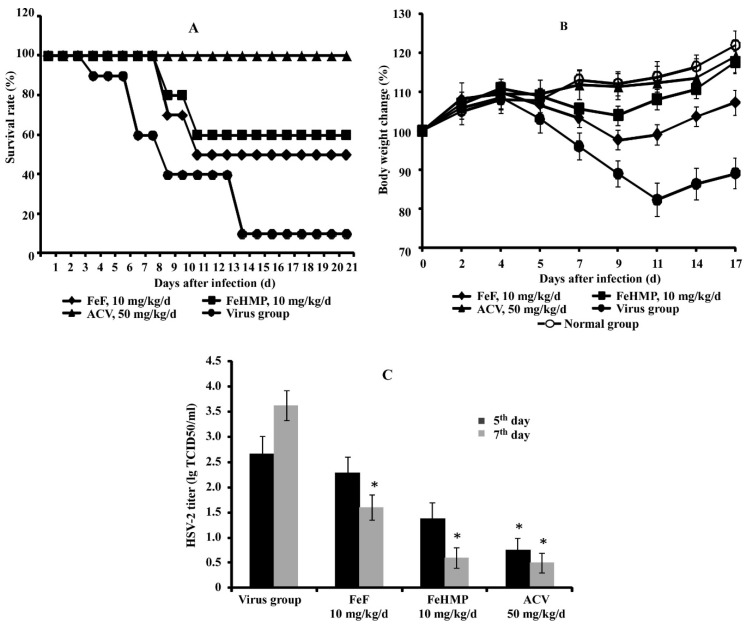
Antiviral activity of fucoidans (FeF and FeHMP) against HSV-2 infection in a mouse vaginitis model. (**A**) Survival analysis of HSV-2-infected and compounds-treated mice; (**B**) body weight change of mice after HSV-2 infection; (**C**) replication of HSV-2 after compound treatment in the mice vaginal lavages at 5 and 7 days after HSV-2 inoculation. * Significant difference between values of the treated group and virus group (*p* ≤ 0.05).

**Table 1 marinedrugs-18-00224-t001:** Spectrum of antiviral activity of fucoidans.

Viruses	Compounds	Pretreatment of Virus		Pretreatment of Cells		Simultaneous Treatment		Treatment of Infected Cells	
		IC_50_ (μg/mL)	SI	IC_50_ (μg/mL)	SI	IC_50_ (μg/mL)	SI	IC_50_ (μg/mL)	SI
**HSV-1**	FeF	106 ± 13	19 ± 2	53 ± 7	38 ± 4	59 ± 8	34 ± 4	80 ± 9	25 ± 3
	FeHMP	127 ± 15	16 ± 2	100 ± 15 *	20 ± 2 *	95 ± 12 *	21 ± 2 *	100 ± 13	20 ± 2
	Acyclovir	NA		NA		2.1 ± 0.3	˃950	0.1 ± 0.01	˃20.000
**HSV-2**	FeF	95 ± 10	21 ± 2	45 ± 6	44 ± 5	50 ± 7	40 ± 5	65 ± 8	31 ± 4
	FeHMP	110 ± 13	18 ± 2	77 ± 8 *	26 ± 3 *	80 ± 10 *	25 ± 3 *	85 ± 11	24 ± 3
	Acyclovir	NA		NA		1.6 ± 0.2	˃1200	0.1 ± 0.01	˃20.000
**ECHO-1**	FeF	710 ± 80	2.8 ± 0.2	105 ± 12	19 ± 2	90 ± 12	22 ± 2	110 ± 13	18 ± 1
	FeHMP	580 ± 65	3.4 ± 0.2	83 ± 10	24 ± 2	85 ± 11	24 ± 2	93 ± 11	21 ± 2
	Ribavirin	NA		NA		˃500	≤1.5	˃500	≤1.5
**HIV-1**	FeF	25 ± 3	8 ± 1	25 ± 3	8 ± 1	25 ± 3	8 ± 1	25 ± 3	8 ± 1
	FeHMP	25 ± 3	8 ± 1	25 ± 3	8 ± 1	50 ± 6	4 ± 0.5	25 ± 3	8 ± 1
	Retrovir	5 ± 0.6	10 ± 1	NA	NA	1.3 ± 0.2	38 ± 4	2.5 ± 0.3	20 ± 3
	Epivir	1.5 ± 0.2	33 ± 4	NA	NA	1.5 ± 0.2	33 ± 4	1.5 ± 0.2	33 ± 4

Note: Values represent the means ± standard deviations of three or more independent experiments; FeF, native fucoidan from brown alga *F. evanescens*; FeHMP, modified fucoidan fragment; Acyclovir, Ribavirin, Retrovir and Epivir were used as positive controls; IC_50_, concentration that inhibited 50% of virus’s replication; SI, selectivity index (CC_50_/IC_50_); NA, no activity; * Statistically significant differences between tested compounds (FeF and FeHMP) (*p* ≤ 0.05).

**Table 2 marinedrugs-18-00224-t002:** Effect of fucoidans treatment on the protection of mice from intravaginal herpes simplex virus 2 (HSV-2) infection.

Groups	Survivors/Total	Protection Index (%)	Average Survival Time (d)
FeF (5 mg/kg/d)	2/10	11.1	11.0 ± 2.1
FeF (10 mg/kg/d)	5/10 *	44.4	14.1 ± 1.4 *
FeHMP (5 mg/kg/d)	3/10	22.2	13.4 ± 1.8
FeHMP (10 mg/kg/d)	6/10 *	55.6	16.0 ± 1.3 *
Acyclovir (50 mg/kg/d)	10/10 *	100	21.0 ± 0.5 *
Virus group	1/10	-	9.7 ± 2.6
Normal group	10/10	100	˃21

Note: Protection index: (Mv − Mc) / Mv ×100%, where Mv and Mc are mortality (%) in the virus and compounds-treated groups, respectively. The tested drugs were administered intraperitoneally for 5 days postinfection. * Doses of tested fucoidans (FeF and FeHMP) were comparable with doses of fucoidans from other brown algae used in experimental viral infections [23,24]. Statistically significant differences between values in the compounds-treated group and virus group (*p* ≤ 0.05).

**Table 3 marinedrugs-18-00224-t003:** The structure of native (FeF) and modified (FeHMP) fucoidans from *Fucus evanescens* [19].

Polysaccharide	Molecular Weight, kDa	OSO_3_^−^ %	Monosaccharide Composition		
			Fuc	Gal	Xyl
FeF	160	28	0.9	0.1	0
FeHMP	50.8	40	1.0	0	0

Antivirals used as positive controls: - Acyclovir^®^, freeze-dried powder for injections (GlaxoSmithKline Manufacturing, Italy) used for herpes virus infections; - Ribavirin^®^ (Sigma-Aldrich, USA) used for enterovirus infections; and - Retrovir^®^ (ViiV Healthcare UK Limited, UK), Epivir^®^ (ViiV Healthcare, UK) used for HIV infection.

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
