# Peer review of "The Comparative Analysis of Antiviral Activity of Native and Modified Fucoidans from Brown Algae Fucus evanescens In Vitro and In Vivo"

_marinedrugs, 2020, doi:10.3390/md18040224_

Round 1

Reviewer 1 Report

The introduction of this paper must be extensively improved. The authors do not offer upfront enough information regarding the structural features of the native and depolymerized version of the fucoidan. Only in section 4.3, that such information, even not in terms of data, is presented. At the beginning of the paper, abbreviations and details are given before the actual explanation, What FeHMP stands for? The explanation is found only in section 4.3 The authors should bring this information to the beginning of the paper and elaborate well the narrative. Moreover, the authors must provide real data and results regarding the NMR spectroscopy and size analyses of both materials rather than just comment on section 4.3.

Author Response

Response:

Dear Reviewer. Thank You very much for Your remarks.

We include all information demanded by You to the text of our paper, including coorections in the introduction, structural features of the native and depolymerized version of the fucoidans, the description of used abbreviations, and NMR-spectroscopy results are provided as supplementary data.

Reviewer 2 Report

The paper by Krylova et al. provide new interesting results about antiviral activity of native and modified fucoidans from brown algae Fucus evanescens in vitro and in vivo. The paper fit with journal scope and and will certainly attract the interest of readers. After close evaluation of manuscript I would suggest revision according next points:

  1. In Introduction I would suggest to cite the recent papers about pharmacokinetic of fucoidans (https://doi.org/10.3390/md16040132; https://doi.org/10.3390/md17120687;) and adjuvant activity (https://doi.org/10.3390/biom9080340).
  2. In Section 2.1. please explain abbreviation CPE
  3. In Section 2.1 (lines 69-70) please clarify which ‘different stages of viruses …replication’ you mind?
  4. In Table 1 please present units for IC50.
  5. In lines 101-102 please clarify which time point you mind under ‘during the early stage of infection’?
  6. In lines after 108-109 please clarify which time point you mind under ‘virus adsorption and penetration to cells (treatment of infected cells)’.
  7. For Table 2 please provide the rationality of doses selection for FeF and FeHMP.
  8. The Fig 1 could be provided in color. I would suggest to use San serif font for this figure to have better resolution.
  9. Section 4.3 – have authors isolated fucoidan from Fucus evanescens or used commercial sample. Please describe shortly isolation procedure.

Author Response

Dear Reviewer. Thank You very much for Your remarks.

Response to point 1: We have included abovementioned citations in the text of "Introduction" section.

Response to point 2: CPE - Cytopathic effect, the correction was made.

Response to point 3: We have clarified and changed the "virus replication" to the "virus infection".

Response to point 4: The correction was made.

Response to point 5: The correction in the text was made: the early stage of infection means "just after the virus inoculation (0 h).

Response to point 6: The correction in the text was made: at 1 h post infection.

Response to point 7: Included.

Response to point 8: The Figure 1 was changed with the help of other graphical editor to improve brightness and legibility

Response to point 9: The description of isolation procedure was added.

Round 2

Reviewer 1 Report

The authors have addressed the concerns previously raised. This manuscript can be accepted for publication as is.

Reviewer 2 Report

The mansucript was revised according my comments and recommendations and it could be accepted in present form.